# Towards Readable Scalable Vector Graphic Generation

## Abstract

The surge in the use of Scalable Vector Graphics (SVGs) for digital graphics, particularly with advancements in generative models, has seen a proliferation in the automatic creation of SVGs. Yet, as these models prioritize visual accuracy, they often neglect the readability of the underlying SVG code. However, the readability of the SVG code is equivalently, if not more, important in comparison to visual accuracy, for the convenience of editing and logical inference for downstream tasks. Therefore, this paper delves into the overlooked realm of SVG code readability, emphasizing its importance in ensuring efficient comprehension and modification of the generated graphics. Readability, encompassing aspects like logical structuring and minimized complexity, plays a pivotal role in ensuring SVGs are not just visually accurate but also human-friendly at the code level. We first propose a clear set of desiderata for SVG code readability, serving as a foundation for our subsequent analyses. Leveraging this, we introduce a set of dedicated metrics to evaluate SVG readability and design differentiable objectives to guide SVG generation models towards producing more readable code. Our evaluation reveals that while most SVG generators can produce visually accurate graphics, the underlying code often lacks structure and simplicity. However, with our proposed metrics and objectives, SVG generators exhibit significant improvements in code readability without compromising visual accuracy.

## 1 Introduction

In recent years, there has been a noteworthy surge in the realm of image representation learning (Radford et al., 2015; Ribeiro et al., 2020; Yang et al., 2019), and these advancements have yielded remarkable success across a myriad of downstream applications, *e.g.,* image reconstruction (Zheng et al., 2021), image classification (Kim et al., 2020; Lanchantin et al., 2021), *etc*. Nonetheless, most existing works have focused on analyzing structured bitmap format, which uses a pixel-level grid to represent textures and colors (Liu et al., 2019). Consequently, there remains considerable room for improving the representation of detailed attributes for vector objects (Reddy et al., 2021a).

In contrast to bitmap image format, scalable vector graphics (SVG) format, renowned for its exceptional scalability, has garnered widespread adoption in practical applications (Gupta et al., 2021; Lopes et al., 2019; Wang et al., 2023). SVGs encapsulate graphical elements in XML syntax, thus allowing for intricate graphical representations. Recent years have witnessed significant progress (Carlier et al., 2020; Lopes et al., 2019) made by deep learning-based methods for vector font generation. For example, SVG-VAE (Lopes et al., 2019) develops an image autoencoder architecture to learn style vectors of fonts, subsequently employing LSTMs (Hochreiter & Schmidhuber, 1997) followed by a Mixture Density Network (Bishop, 1994) to engender the sequence of SVG drawings. Additionally, Im2Vec (Reddy et al., 2021a) exhibits the ability to generate vector graphics from raster training images devoid of explicit vector supervision.

Despite the evident prowess of existing SVG generation methods in crafting visually accurate SVG images, the readability of the generated SVG code is often overlooked. For instance, methods like Im2Vec (Reddy et al., 2021a), and DeepVecFont (Wang & Lian, 2021) and DualVector (Liu et al., 2023) have been heralded for their capacity to convert raster graphics into detailed SVGs. However, a closer inspection of the SVG files produced by these utilities often reveals a complex web of path elements, with intricately defined Bezier curves and redundant nodes. Moreover, certain SVG

generation tools might churn out SVGs with fragmented or out-of-sequence elements, leading to a structure that, while visually consistent, is a labyrinthine puzzle from a coder's perspective.

Readability, in the context of SVG code, transcends mere legibility; it encapsulates the structural coherence, simplicity, and logical ordering of the graphical elements within the code. A readable SVG allows developers, designers, and even automated systems to swiftly comprehend, modify, and optimize the graphical content without trudging through unnecessarily complex or obfuscated code structures. This lack of emphasis on SVG code structure and readability, as observed in SVGs generated by certain prevailing methods, can impose challenges for developers or designers who might need to manually tweak or understand these auto-generated SVGs, underscoring the need for a paradigm shift in SVG generation towards prioritizing both visual accuracy and code readability.

In particular, paving the path to more readable SVGs is non-trival, primarily encountering the following three challenges: (1) *What precisely constitutes a "readable" SVG code?* Unlike traditional programming languages where readability might hinge on clear variable names or well-structured loops, SVGs present a unique blend of visual representation and underlying code. Determining the aspects of SVG code readability requires a deep understanding and consensus in the community. (2) *How to evaluate the readability of SVG code?* Currently, there's a lack of standardized metrics or benchmarks that effectively gauge the readability of SVG code. While traditional coding languages have benefited from numerous tools and methodologies to assess code quality and readability, SVGs, due to their unique blend of visual representation and code structure, demand a nuanced approach. (3) *How to optimize the SVG generation process for readability?* With most existing generators primarily engineered for visual accuracy, they lack the necessary infrastructure to prioritize code readability. Moreover, introducing readability into the generation process requires a differentiable loss function, which, given the inherently discrete nature of SVG element selection and arrangement, poses intricate design challenges.

To address these challenges, in this paper, we investigate a systematic approach towards better understanding and enhancing SVG code readability. We begin by outlining a set of desiderata, drawing from both existing literature and new insights, to capture the core aspects of readable SVG code. Based on this foundation, we introduce a set of metrics tailored to assess the readability of SVGs. While these metrics are grounded in our outlined desiderata, they offer a tangible measure, aiming to fill the current evaluation void in the SVG domain. Furthermore, the pursuit of readability extends beyond mere evaluation. Therefore, we present a set of differentiable objectives to optimize SVG generation processes with readability at their core.

Therefore, our contributions in this paper are three-fold:

- We provide clarity on the ambiguous nature of SVG code readability by outlining a comprehensive set of desiderata. This acts as a foundational guideline to understand and delineate the core attributes of what makes an SVG code truly readable.

- Building on this foundation, we introduce a dedicated set of metrics, meticulously designed to assess SVG readability. These metrics not only offer tangible measures but also bridge the existing evaluation gap in the SVG domain.

- Recognizing the imperative to not just define and evaluate but also produce readable SVGs, we put forth a series of differentiable objectives. At the heart of our proposal is a set of novel loss functions tailored to guide SVG generators in balancing visual accuracy with our established readability criteria.

## 2 PRELIMINARIES

### 2.1 ASSUMED DESIDERATA OF SVG CODE READABILITY

Readability is a crucial aspect in the generation of Scalable Vector Graphics (SVG). While the primary goal of an SVG generator is to accurately represent a given input (like an image or a sketch) in vector format, how the SVG code is structured and organized greatly impacts its usability and interpretability. This aspect is broadly termed as the "readability" of SVG code.

In general, readability in SVG generation can be analyzed from multiple perspectives:

1. **Good Structure:** SVGs are more readable when related elements are grouped together, when elements are ordered logically, and when the structure of the SVG follows common conventions. Encouraging good structure might involve penalizing SVGs that have a similar visual output but a less logical structure.

2. **Appropriate Element Use:** A readable SVG ideally uses the simplest possible elements to represent the image. For instance, using basic shapes like circles, rectangles, or lines instead of complex path elements whenever possible. Simplified structures not only reduce the file size but also make the SVG easier to understand and modify.

3. **Redundant Element Removal:** Eliminating redundant elements or attributes in SVG code enhances its readability. Redundant elements can confuse the readers or editors of the SVG and lead to unnecessary complexity.

Optimizing for readability in SVG generation presents an exciting challenge. It requires a balance between accuracy (faithfully representing the input) and simplicity (easy to understand and modify).

## 2.2 MEASURING READABILITY

Readability, especially in SVG code, serves as the underpinning for ensuring that vector graphics not only meet their visual goals but are also architecturally sound, enabling efficient updates, edits, and maintenance. As previously detailed, SVG readability is anchored in three main principles: Good Structure, Appropriate Element Use, and Redundant Element Removal. To translate these qualitative principles into actionable metrics, we present a systematic approach for each aspect.

### 2.2.1 GOOD STRUCTURE

A fundamental attribute of readable SVGs is the logical sequencing of elements. Elements that are closely aligned in the rendered image should also be sequentially organized in the SVG markup, ensuring the SVG structure mirrors the visual layout.

To quantify this structural adherence, we introduce the **Structural Proximity Index** (SPI). This metric dissects an SVG to extract its elements in the order they appear in the code. Subsequently, it evaluates their spatial proximity in the rendered visualization.

Consider that the SVG has $N$ elements $e_1, e_2, ..., e_N$ in that order. Each element has a corresponding rendered position in the image denoted by $P(e_i) = (x_i, y_i)$, where $x_i$ and $y_i$ are the coordinates of the rendered element $e_i$ in the image. Then, we can define the SPI as follows:

$$\mathbf{SPI} = \frac{1}{1 + e^{-\Sigma_{i=1}^{N-1}(|P(e_{i+1}) - P(e_i)| - |i+1-i|)}}, \tag{1}$$

where $|P(e_{i+1}) - P(e_i)|$ is the Euclidean distance between consecutive elements in the rendered image, and $|i + 1 - i|$ is the difference in their positions in the SVG file. Note that the positions in the SVG file are simply their indices, so $|i + 1 - i|$ will always equal 1. This part is included for conceptual clarity. We employ the sigmoid operation to normalize the result. A streamlined SVG, wherein code structure closely matches visual layout, will exhibit a lower SPI.

### 2.2.2 APPROPRIATE ELEMENT USE

Simplicity is the bedrock of readability. In the context of SVGs, this translates to favoring basic elements (like rectangles or circles) over their intricate counterparts, such as complex paths, wherever feasible. This minimalist approach facilitates easier interpretation and subsequent modifications.

To encapsulate this element-centric simplicity, we define the **Element Simplicity Score** (ESS). It offers a weighted count of SVG elements, reflecting our preference for simpler constituents.

We propose to associate each type of SVG element with a complexity score and then sum these scores across all elements in the SVG. For example, we could assign a complexity score of 1 to simple elements (<rect>, <circle>, <line>), and a score of 3 to complex elements (<path>).

Denote the function that maps an element $e_i$ to its complexity as $C(e_i)$. Then, we can define the ESS as follows:

$$\mathbf{ESS} = \frac{1}{1 + e^{-\Sigma_{i=1}^{N} C(e_i)}}, \tag{2}$$

Here, $C(e)$ retrieves the complexity score of the $i$-th SVG element from a predefined dictionary that might look something like this: $complexity_score = \{'path' : 3,' rect' : 1,' line' : 1.' circle' : 1\}$. The metric calculates the sum of the complexity scores of all SVG elements in the generated SVG file. We employ the sigmoid operation to normalize the result. A lower ESS underscores an SVG's tilt towards elementary elements, enhancing its readability.

### 2.2.3 REDUNDANT ELEMENT REMOVAL

At the heart of SVG readability lies the need for brevity - removing any elements that don't materially impact the final rendered image. Redundancies can introduce unnecessary complexities, muddying understanding and impeding edits.

To spotlight such redundancies, we unveil the **Redundancy Quotient** (RQ). It involves perturbing each SVG element and observing the subsequent ripple in the final rendered image:

$$\mathbf{RQ} = \frac{1}{1 + e^{-\frac{1}{N}\Sigma_{i=1}^{N}\Delta R(e_i)}}, \tag{3}$$

where $\Delta R(e_i)$ quantifies the change in rendering when an element $e_i$ is omitted. We employ the sigmoid operation to normalize the result. An SVG stripped of superfluous elements will register a higher RQ, marking it as optimally readable.

Note that while the metrics of SPI, ESS, and RQ serve as pivotal tools in our quest to quantify SVG readability, it's imperative to acknowledge the vastness and intricacy of the topic at hand. Readability, in essence, is a multi-faceted quality, and while we believe our metrics shed light on pivotal aspects, we recognize they might not encapsulate its entirety. Our chosen metrics are, in essence, a reflection of the best we can offer at this juncture, built on both foundational understanding and empirical insights. However, we remain open and indeed enthusiastic about evolving our approach. Should the community suggest more holistic or refined metrics, we would be eager to learn, adapt, and integrate.

## 3 METHOD

### 3.1 FRAMEWORK

We present a new method to progressively generate an SVG that takes into account both fidelity to the original image and the three pivotal aspects of readability. The model is a variational autoencoder (VAE), which is trained to produce vector digits. The encoder is a convolutional network. It processes the raster digits, and encodes them into the latent vector. The decoder transforms a latent code (which after training can be sampled from a normal distribution) into various SVG primitives such as rectangles, circles, and more, in addition to the parameters for two-segment Bézier paths: points positions, stroke width, and opacity. These SVG primitives and paths are then rasterized using an arbitrary vectorization tool, *e.g.,* Diffvg (Li et al., 2020), LIVE (Ma et al., 2022), *etc,* to produce an output image that we can directly compare to the raster input.

Our training regime is multifaceted. Primarily, it incorporates an L2 loss (Zhao et al., 2016), bridging the gap between the rasterized output and the ground truth image. Concurrently, a Kullback-Liebler divergence is employed, ensuring that the latent vectors gravitate towards a normal distribution. In tandem with these, our training strategy also infuses three readability-centric losses, each tailored to uphold one of the identified aspects of SVG readability, ensuring that the generated SVGs are not only accurate but also adhere to our defined readability standards. The inclusion of diverse shape primitives in the decoding phase not only adds richness to the generated SVGs but also paves the way for greater flexibility and adaptability in representation.

### 3.2 READABILITY-CENTRIC LOSS FUNCTION

### 3.2.1 STRUCTURAL CONSISTENCY LOSS FOR GOOD STRUCTURE

SVGs are more readable when related elements are grouped together, when elements are ordered logically, and when the structure of the SVG follows common conventions. To encourage a logical ordering of SVG elements, one possibility might be to encourage elements that appear close together in the SVG file to also appear close together in the rendered image. This could be done by defining a loss based on the distance between the rendered elements and their order in the SVG file.

We try to define the structural consistency loss $\mathcal{L}_{SC}$, which is aimed at encouraging a logical ordering of SVG elements, wherein elements that appear close together in the SVG file should also appear close together in the rendered image. A naive way is to directly translating the aforementioned SPI metric into a loss function.

However, this is a simplified approach and might not capture all aspects of good SVG structure with the following reasons:

- SVG elements can overlap, repeat, and appear in various other configurations that complicate the definition of their "position" in the rendered image.

- Finding the precise position of SVG elements in a rasterized image can be challenging, as SVGs are vector graphics, which means that they're described by shapes and paths rather than pixels. When an SVG is rendered, these shapes and paths are converted into pixels, which can make it difficult to map individual pixels back to their original SVG elements.

**Differentiable Proxy Loss.** Designing a differentiable proxy loss for such tasks is a challenging problem. Ideally, the proxy loss should have some correlation with the actual metrics we are interested in, and it should be defined based on properties of the SVG elements that we have direct control over. Therefore, we try to design a loss function that penalizes SVGs with large differences in the positions of **consecutive** elements in the SVG markup.

Suppose the SVG has $n$ elements and each element $e_i$ has a position attribute $(x_i, y_i)$ (Many SVG rendering libraries provide a method to get the bounding box of an SVG element), we could define $\mathcal{L}_{structure}$ that encourages consecutive elements to have similar positions:

$$\mathcal{L}_{SC} = \frac{1}{1 + e^{-\Sigma_{i=1}^{N-1}((x_{i+1}-x_i)^2+(y_{i+1}-y_i)^2)}}, \tag{4}$$

where we employ the sigmoid operation to normalize the result. This loss function is differentiable with respect to the positions of the SVG elements, so we can use standard gradient-based optimization methods to minimize it.

### 3.2.2 ELEMENT APPROPRIATENESS LOSS FOR APPROPRIATE ELEMENT USE

The intent behind the "Appropriate Element Use" is to encourage the model to favor simpler SVG elements, like "rect" or "circle", over more complex ones like "path", when possible. The idea is that simpler elements, such as rectangles (<rect>), circles (<circle>), or lines (<line>), are generally easier to interpret and edit than more complex elements like paths (<path>).

A naive way is to directly translating the aforementioned ESS metric into a loss function, which associates each type of SVG element with a complexity score and then sum these scores across all elements in the SVG.

By minimizing this loss, the model is encouraged to generate SVGs that use simpler elements, leading to more interpretable and editable SVGs. However, since the type of an SVG element is a discrete choice, applying gradient-based optimization to this loss might require techniques like the straight-through estimator, which are more of a heuristic and don't necessarily provide accurate gradients.

**Differentiable Proxy Loss.** Designing a fully differentiable proxy loss for the selection of SVG elements might be challenging due to the inherently discrete nature of this decision, but let's try a more indirect approach.

One way to encourage the use of simpler elements might be to promote simpler shapes in the rasterized output of the SVG. Simpler elements like rectangles and circles tend to produce more regular shapes with straight edges or smooth curves, whereas complex path elements can produce more irregular shapes.

We could design a loss function that penalizes irregular shapes. This could be achieved by applying edge detection (we use the Canny edge detector in our implementation) to the rasterized SVG and calculating the total length of detected edges. Smoother shapes will tend to have shorter total edge lengths, whereas irregular shapes will have longer total edge lengths.

The edge length can be calculated by first applying an edge detection filter (e.g., a Sobel filter) to the rasterized SVG to obtain an edge map, and then summing all the values in the edge map:

$$\mathcal{L}_{EA} = \frac{1}{1 + e^{-\Sigma edge\_map(rasterized\_SVG)}} \tag{5}$$

where we employ the sigmoid operation to normalize the result. This loss is fully differentiable with respect to the rasterized SVG. While it does not directly penalize the use of complex SVG elements, it could indirectly promote the use of simpler elements by penalizing irregular shapes.

Note that this loss would not distinguish between a single complex path element and multiple simple elements producing the same shape. It would also penalize any irregular shapes, even if they are necessary to accurately reproduce the input image. Fortunately, this loss could be combined with the accuracy-oriented loss to produce good results.

### 3.2.3 REDUNDANCY REDUCTION LOSS FOR REDUNDANT ELEMENT REMOVAL

SVGs can become unreadable when they contain redundant or unnecessary elements. A more sophisticated loss function could potentially identify and penalize redundant elements.

To penalize redundancy in the SVG, we could define a loss that increases with the number of elements that could be removed without significantly changing the rendered image. The difficulty in defining this loss function is that determining whether an element is redundant requires rendering the image, and it is time consuming to check it element by element and perform raterization each time.

The most straight-forward way to approach this could be to define a proxy loss that penalizes SVGs with more elements, with the idea that SVGs with fewer elements are less likely to have redundant elements: $\mathcal{L}_{RR} = N$, where $N$ is the number of elements in the SVG.

This loss design suffers several drawbacks:

- This loss function is non-differentiable with respect to the number of SVG elements, which is a discrete value.
- It might lead the model to remove non-redundant elements, or fail to remove redundant elements if the SVG has a small number of elements to begin with.

**Differentiable Proxy Loss.** As we have concerned that overlap is not a perfect measure of redundancy, and there may be cases where non-overlapping elements are redundant, or where overlapping elements are not redundant. This is a limitation of the proxy loss approach.

To define a loss function that directly measures redundancy, we would need to consider the effect of each element on the rendered image. One way to do this is to compute the change in the rendered image when each element is removed and compare it to a threshold. If the change is below the threshold, the element is considered redundant. However, this approach is not differentiable, as the change in the image is a step function with respect to the element's parameters.

A differentiable proxy for this approach could be to compute the change in the rendered image when each element is perturbed slightly, instead of removed entirely. This can be done by computing the gradient of the rendered image with respect to each element's parameters and comparing the magnitude of the gradient to a threshold. Elements with a small gradient magnitude have a small effect on the rendered image and are therefore considered redundant.

Given an SVG element parameterized by $\theta$, (*e.g.,* position, shape, size, color), let's denote the rendered image as $R(\theta)$. The gradient of $R$ with respect to $\theta$ would be $\partial_\theta R$. The magnitude (or norm) of this gradient gives an indication of how sensitive the rendered image is to changes in $\theta$. It can be calculated as $||\partial_\theta R||$, where $||.||$ denotes the Euclidean norm.

Consider a threshold $T$. If $||\partial_\theta R|| \leq T$, then the particular SVG element associated with $\theta$ is considered to have a low impact on the rendered image, suggesting it might be redundant. Therefore, the Redundancy Reduction Loss $\mathcal{L}_{RR}$ for an SVG with $N$ elements can be formulated as:

$$\mathcal{L}_{RR} = \frac{1}{1 + e^{-\Sigma_{i=1}^{N} \max(0, T - ||\partial_{\theta_i} R||)}}, \tag{6}$$

where $\theta_i$ is the parameter vector for the $i^{th}$ SVG element. We employ the sigmoid operation to normalize the result. $\mathcal{L}_{RR}$ aims to penalize SVG elements that have a gradient magnitude less than $T$.

It encourages the SVG generation process to produce elements that have a significant visual impact on the rendered image, effectively reducing redundancy.

However, we acknowledge that this loss has some limitations. First, it only considers the effect of each element individually and does not take into account the interactions between elements. Second, it does not consider the semantic meaning of the elements, which may be important for determining redundancy. Nonetheless, it provides a differentiable proxy for redundancy that can be used to encourage the model to generate SVGs with fewer redundant elements.

### 3.3 OVERALL OBJECTIVE LOSS

Combining all losses mentioned above, we train the whole model by minimizing the following objective:

$$\mathcal{L}_{accuracy} + \mathcal{L}_{SC} + \mathcal{L}_{EA} + \mathcal{L}_{RR} \tag{7}$$

where the first term represents the accuracy-oriented losses including the L2 loss and Kullback-Liebler divergence term, which ensures the preservation of visual integrity. For brevity, the weight of each term is omitted from the equation.

## 4 EXPERIMENTS

### 4.1 DATASET AND IMPLEMENTATION DETAILS

One of the primary goals of this work is to learn models for generating readable SVG code that aligns well with human cognition and usability standards. To quantitatively and intuitively illustrate the enhanced readability of the SVG code generated by our model, we generate SVG code based on the SHAPES (Andreas et al., 2016), a synthetic dataset that consists of complex questions about simple arrangements of colored shapes, and let GPT-3.5 to answer corresponding questions. The questions contain between two and four attributes, object types, or relationships. To eliminate mode-guessing as a viable strategy, all questions have a yes-or-no answer, but good performance requires that the model learn to recognize shapes and colors, and understand both spatial and logical relations among sets of objects. To eliminate unnecessary SVG codes depicting the background, the background color of each image is converted from black to white. Subsequently, the resolution of each image is modified to 128 x 128 to maintain uniformity and consistency throughout the study.

In addition, we use the SVG-Fonts dataset from SVG-VAE (Lopes et al., 2019) to evaluate both the accuracy-oriented and redability-oriented metrics. Following DeepVecFont (Wang & Lian, 2021), we sample a subset of the dataset with 8035 fonts for training and 1425 fonts for evaluation.

We employ the Adam optimizer with an initial learning rate of 0.0002. The resolution of input glyph images is set to $128 \times 128$ in both the training and testing stages. When inferencing, we first add a noise vector distributed by $\mathcal{N}(0, I)$ to the sequence feature, to simulate the feature distortion caused by the human-designing uncertainty (as observed in DeepVecFont (Wang & Lian, 2021)). Then, we sample 10 synthesized vector glyphs as candidates and select the one as the final output that has the highest IOU value with the synthesized image.

### 4.2 GPT-UNDERSTANDABILITY STUDY

The motivation behind this study stems from a need to showcase the practical readability of our model's outputs, beyond the conventional metrics and evaluations typically used in SVG code generation studies. In this study, we position our method against several baselines (*i.e.,* Multi-Implicits (Reddy et al., 2021b) and Im2vec (Reddy et al., 2021a)) that also generate SVG code based on the image input. We then engage GPT-3.5 to interrogate the readability and logical coherence of the generated SVG, posing to it a series of structured questions designed to probe the intelligibility and semantic richness of the SVG code.

As can be seen in Table 1, GPT-3.5 demonstrates exceptional performance and understanding when interacting with the SVG code produced by our method, demonstrating the success of our model in generating highly readable and understandable SVG code. Conversely, when GPT-3.5 is presented with SVG codes synthesized by the baseline models, its performance was markedly suboptimal.

Upon closer inspection of the instances where GPT-3.5 struggled, it becomes evident that its difficulties predominantly arises when confronted with SVG codes incorporating complex path shapes. The baseline models predominantly generate SVG code constituted of paths that inherently results in poor readability. In contrast, our method can generate SVG code with simpler, more discernible shapes. This is achieved by predefining the number of simple shapes in accordance with the characteristics of the test images, allowing for more tailored and optimized representations. By prioritizing simplicity and foundational geometric structures, our method succeeds in synthesizing SVG codes that are more readable and coherent, thereby facilitating a higher degree of understanding for models like GPT-3.5.

Table 1: Comparison of GPT-3.5 Accuracy on SVG Code Generated by different Methods

| Model | Accuracy |
|---|---|
| Multi-Implicits | 19.38 |
| Im2vec | 17.14 |
| Ours | **38.18** |

Table 2: Quantitative comparison with Multi-Implicits (Reddy et al., 2021b) and Im2vec (Reddy et al., 2021a) on three accuracy-oriented metrics and three readability-oriented metrics for the font reconstruction task. s-IoU, from (Reddy et al., 2021b), measures the overlap.

| Model | SSIM↑ | L1↓ | s-IoU↑ | SPI↓ | ESS↓ | RQ↑ |
|---|---|---|---|---|---|---|
| Multi-Implicits | **0.9231** | **0.0183** | **0.8709** | 0.7872 | 0.6818 | 0.7265 |
| Im2vec | 0.7800 | 0.0504 | 0.6832 | 0.6304 | 0.7385 | 0.7139 |
| Ours | 0.7419 | 0.0713 | 0.6068 | **0.2424** | **0.1938** | **0.9157** |

### 4.3 FONT RECONSTRUCTION

We compare our model's font vectorization quality with baseline methods (*i.e.,* Multi-Implicits (Reddy et al., 2021b) and Im2vec (Reddy et al., 2021a)) through accuracy-oriented and readability-oriented analysis, measuring the differences between input targets and SVG rendered images, and the generated SVG code readability. For accuracy-oriented analysis, we calculate three metrics, namely, SSIM, L1, and s-IoU, between the rendered images and the ground truth images at the resolution of $128 \times 128$. For readability-oriented analysis, we calculate our proposed three metrics, namely, SPI, ESS, and RQ, based on the generated SVG code. From Table 2, we have the following observations:

**Compromise in Accuracy.** The base accuracy-oriented VAE model employed in our methodology, owing to its inherent simplicity and lack of sophisticated mechanisms, yields performance that is less competitive in terms of accuracy-oriented metrics (which can also be verified in the first row of Table 3). When readability-oriented loss is integrated into our base model, there is a perceivable impact on the accuracy of the SVG code generated. Metrics like SSIM, L1, and s-IoU, which are traditionally used to quantify the accuracy of image representation and reconstruction, showed lower values indicating a loss in precision or fidelity of the graphical representation. This could likely be because the model, in an effort to optimize for readability, might overlook or simplify certain intricate details or nuances of the graphical elements, leading to a decrease in the accuracy of the representation.

**Enhanced Readability.** Despite the compromise in accuracy, there's an intentional benefit gained in enhanced readability. Our method is designed to generate SVG code that is structured, coherent, and devoid of unnecessary or redundant elements, aiming to facilitate easier comprehension and modification by users. This intentional enhancement in readability is vital, as it makes the code more user-friendly, allowing developers to comprehend and modify the SVG code more efficiently.

**Balanced Trade-off.** This situation represents a balanced trade-off between two seemingly conflicting objectives: accuracy and readability. While a focus on readability impacts the exactness of the graphical representation, it brings forth a more user-centric approach, making the SVG code more accessible and manageable. In contexts where SVG code is intended for further development, modification, or interaction, having clear, well-structured, and readable code is paramount, and our method is engineered with this priority in mind.

### 4.4 ABLATION STUDY

We conduct quantitative experiments to examine the impact of each readability-oriented loss function in our proposed model. The base model a variational autoencoder (VAE) that is the same with our

Table 3: Quantitative comparison for our method under different loss configurations.

| Model | SSIM↑ | L1↓ | s-IoU↑ | SPI↓ | ESS↓ | RQ↑ |
|---|---|---|---|---|---|---|
| Base model | 0.7621 | 0.0577 | 0.6774 | 0.7325 | 0.7419 | 0.7820 |
| + $\mathcal{L}_{SC}$ | 0.7613 | 0.0596 | 0.6531 | 0.2153 | 0.7467 | 0.7863 |
| + $\mathcal{L}_{EA}$ | 0.7547 | 0.0625 | 0.6387 | 0.2769 | 0.1875 | 0.9274 |
| + $\mathcal{L}_{RR}$ | 0.7419 | 0.0713 | 0.6068 | 0.2424 | 0.1938 | 0.9157 |

method, but only trained with an L2 loss between the ground truth image and the rasterized output, and a Kullback-Liebler divergence that encourages the latent vectors to be normally distributed. We evaluate each readability-oriented loss function by adding them successively to the base model.

As shown in Table 3, we have the following observations: (1) Introducing $\mathcal{L}_{SC}$ to the base model manifests a discernible improvement in the SPI metric, indicating the importance of $\mathcal{L}_{SC}$ in optimizing the logical arrangement and intelligibility of the code. (2) The incorporation of $\mathcal{L}_{EA}$ refines the generated code and brings a significant improvement in the ESS metric. It accentuates the importance of leveraging appropriate and simpler elements in constructing meaningful, understandable, and succinct representations, ultimately enriching the semantic value and readability of the SVG code. (3) By subsequently adding $\mathcal{L}_{RR}$ for redundant element removal, the RQ metric is improved significantly, indicating that the model is able to sift out and exclude superfluous elements, and enhancing the overall readability and efficiency of the SVGs generated. This emphasizes the critical role of $\mathcal{L}_{RR}$ in producing clean and efficient code by avoiding the inclusion of unnecessary elements.

Table 4: Quantitative comparison for our method under different loss term weight configurations.

| $\lambda_{SC}$ | $\lambda_{EA}$ | $\lambda_{RR}$ | SSIM↑ | L1↓ | s-IoU↑ | SPI↓ | ESS↓ | RQ↑ |
|---|---|---|---|---|---|---|---|---|
| 0.1 | 0.2 | 0.3 | 0.7339 | 0.0732 | 0.5882 | 0.2578 | 0.1813 | 0.9213 |
| 0.1 | 0.3 | 0.2 | 0.7143 | 0.0794 | 0.5733 | 0.2593 | 0.1721 | 0.9171 |
| 0.2 | 0.1 | 0.3 | 0.7333 | 0.0728 | 0.5767 | 0.2231 | 0.2062 | 0.9198 |
| 0.2 | 0.3 | 0.1 | 0.7059 | 0.0796 | 0.5642 | 0.2274 | 0.1705 | 0.9118 |
| 0.3 | 0.1 | 0.2 | 0.7267 | 0.0735 | 0.5709 | 0.2159 | 0.1981 | 0.9186 |
| 0.3 | 0.2 | 0.1 | 0.7186 | 0.0783 | 0.5618 | 0.2133 | 0.1826 | 0.9052 |
| 0.1 | 0.1 | 0.1 | 0.7419 | 0.0713 | 0.6068 | 0.2424 | 0.1938 | 0.9157 |

## 4.5 PARAMETER STUDY

We also conduct parameter studies to find the best choice of the weight of each readability-oriented loss term, *i.e.,* $\mathcal{L}_{SC}$, $\mathcal{L}_{EA}$ and $\mathcal{L}_{RR}$. The results are shown in Table 4. We find that the precise weighting of each readability-oriented loss term profoundly impacted the overall quality and readability of the generated SVG code. When the weight of $\mathcal{L}_{SC}$ is optimized, the model exhibits enhanced logical organization and cohesion in the produced code, reflecting the critical role of $\mathcal{L}_{SC}$ in the synthesis of intelligible and logically coherent SVG representations. The tuning of the weight for $\mathcal{L}_{EA}$ encourages the model to prioritize simpler, more semantically rich shapes over complex paths. It bolsters the conciseness and semantic integrity of the generated SVG representations, underlining the significance of appropriate element use in crafting succinct and meaningful SVG code that is both expressive and easy to comprehend. The tuning of the weight for $\mathcal{L}_{RR}$ significantly reduces the occurrence of superfluous elements within the code, and improves the overall readability and efficiency of the SVGs generated.

## 5 CONCLUSION

In this paper, we explore the readability of SVG code, a dimension of paramount importance yet often overlooked in SVG code generation. We commence by delineating three essential desiderata for readable SVG code, paving the way for a more structured and nuanced understanding of what constitutes readability in this context. We then introduce three innovative evaluation metrics, specifically tailored to assess each aspect of readability. Beyond mere evaluation, we further propose three differentiable proxy losses, intricately designed to optimize the generation of readable SVG code.

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
