# OpenReview forum: "Towards Readable Scalable Vector Graphic Generation"
_ICLR.cc/2024/Conference — Submitted to ICLR 2024_

### Official Review · Reviewer_z767 · 2023-10-27

**Soundness:** 1 poor
**Presentation:** 3 good
**Contribution:** 3 good
**Rating:** 5
**Confidence:** 5

**Summary:**

This paper examined the challenge of having good code readability for generated SVG. The idea itself is novel and usually overlooked by earlier works. Authors proposed three metrics for measuring readability and design their corresponding differentiable proxy losses. Evaluation results on font and simple shape data show improvements on the code readability metric when using the authors’ method.

**Strengths:**

Code readability for DSL generation like SVG or CAD are often overlooked by existing work. This paper addresses a very important challenge and the proposed evaluation metrics are more or less reflective of a good readable code. Authors not only contribute to designing the metrics but also the proxy losses for incorporating them into the optimization pipeline. Both the task formulation and solution are novel.

**Weaknesses:**

Many details and supportive proof are missing. This significantly impacts the soundness.

For example, paper provides no visual results, making it very hard to judge the quality of the method. Many terms used throughout the paper also need to be explained more. Just listing a few: In equation 1, ‘x and y are the coordinates of the rendered element’. So how is this coordinate decided, is it the middle point or start/end? Equation 2, ‘complex elements’, so what is the definition for ‘complex’? Is it based on the number of basis functions or the number of knots? Equation 3 ‘it involves perturbing each SVG…’, how is this perturbing conducted? Section 3.1 ‘various SVG primitives such as rectangles, circles, and more’,  can you write out all the types?

Also the experiments are conducted on simple shapes and digits, there are large scale SVG dataset like the one used in IconShop. If authors tested their metrics on a more general setting this would make it more supportive. They also mentioned that reconstruction quality will decrease. I wonder how bad this is? Again, I can not evaluate since there are no visual results.

**Questions:**

While I like the direction of this paper and the proposed metrics and solution are interesting, it currently suffers from lack of soundness (See what I write in weakness). Authors should at least provide some visual results.

---

### Official Review · Reviewer_p9Qj · 2023-10-29

**Soundness:** 2 fair
**Presentation:** 2 fair
**Contribution:** 2 fair
**Rating:** 5
**Confidence:** 3

**Summary:**

The paper studies the problem of scalable vector graphics (SVG) generation, with a particular focus on improving the readability of generated SVG code.

The main contribution is to propose a set of quantitative metrics to evaluate SVG code readability in terms of  structure, simplicity and redundancy,  and design corresponding training objectives to guide models to generate readable SVG code.

**Strengths:**

1. Improving the code readability of generated SVGs, an crucial yet under-explored aspect in SVG generation, is an important and meaningful attempt, which can make model-generated SVGs more readily usable in practical applications.

2. The proposed training objectives are shown to be effective in improving code readability through the experiments.

**Weaknesses:**

1. The design choices of some metrics are questionable. First, SPI in Sec. 2.2.1 uses the Euclidean distance between the center positions of two elements as their spatial proximity, which will not work well when one or two of them are quite large. Suppose there are two circles that touch each other. Their perceptual distance is zero, but their center distance is rather large due to their large sizes. Thus, the center distance may not be a proper measure here, and better choices need to be explored, e.g., the minimum distance between two elements. Second, ESS in Sec. 2.2.2 sets score 1 and 2 to simple and complex elements, respectively, but these score values are selected without any theoretical or experimental justification.

2. The design choice of the element appropriateness loss is inadequate. The loss uses the edge length of an element to measure its complexity, which is problematic. While such a loss can penalize complex elements with irregular shapes, it will also give a large penalty score to large simple elements, which also have long edges.

3. A thorough study on accuracy-readability trade-off is missing. As shown in Table 2, accuracy is traded for readability. Given the results, it is natural to ask if it is possible (and how) to find an optimal point between accuracy and readability with the proposed readability objectives. A study should be performed to investigate this question, e.g., by introducing a weight to the accuracy loss and plotting the accuracy and readability scores with respect to various settings of the weight.

**Questions:**

In Eq.(3), how is the quantity for rendering change computed exactly?

---

### Official Review · Reviewer_pKgQ · 2023-11-01

**Soundness:** 1 poor
**Presentation:** 2 fair
**Contribution:** 1 poor
**Rating:** 3
**Confidence:** 5

**Summary:**

The goal of this paper is to improve the readability of SVGs generated by generative models. The paper proposed three metrics to measure the readability of SVGs: (1) Euclidean distance between consecutive SVG elements, (2) number of elementary elements, (3) number of redundant elements. The paper also proposed three losses to improve the readability of SVGs: (1) penalize large distance between consecutive SVG elements, (2) penalize the sum of the intensity of the edge map, (3) penalize redundant elements. The experiments are done on a synthetic dataset and a vector font dataset.

**Strengths:**

The discussion on evaluating the editability of generated SVGs is interesting, which is an important topic for future research on SVG generation.

**Weaknesses:**

- The motivation for improving the readability of SVGs is lacking. For most SVGs, it doesn't make sense to "read" them, especially for those large SVGs.
- A more important property of SVGs is editability. However, the metrics proposed in the paper are lacking for measuring editability. For example, a small distance between consecutive SVG elements doesn't mean good editability. Also, representing a complex shape with a lot of small rectangular shapes has bad editability but can still achieve good values in all three metrics proposed in the paper.
- Implementation details are missing. For example, what's the architecture of the SVG decoder?
- In Table 2, except for the proposed metrics, all other metrics are worse. It will be helpful to show some qualitative results.

**Questions:**

Why is 3 chosen for the complexity score of path?

---

### Official Review · Reviewer_Zk7q · 2023-11-09

**Soundness:** 2 fair
**Presentation:** 1 poor
**Contribution:** 2 fair
**Rating:** 3
**Confidence:** 4

**Summary:**

This paper aims to improve the readability for generated scalable vector graphic (SVG) drawings. The authors provide three aspects to define readability for SVG files: good structure, appropriate element use, and redundant element removal. Based on these aspects, the authors propose three corresponding metrics that measure these aspects. Finally, they design differentiable proxy losses for each metric and apply them to an existing model for SVG generation. The experiments show that the model learns to better balance between the reconstruction accuracy and the proposed metrics compared to baseline models.

**Strengths:**

- Evaluating and improving generative models for different modalities and use cases is an important research direction, and this paper addresses one of the neglected areas in this direction: the readability for SVG generation.
- The authors propose new criteria and metrics to define and evaluate the readability of SVG files.
- The designed differentiable proxy losses can be potentially applied to different types of generative models.

**Weaknesses:**

- No qualitative results or visualizations provided. It will not only help readers to better understand the criteria, but also demonstrate how the proposed method improve the SVG generation compared to the baseline methods.

- User study can be very useful. For example, let the participants pick the more readable one from pairs of (original, generated) rendered SVG images.

- Lack of details on the model. There is not enough detail provided to the model used in the experiments. The formulation/representation of the vector graphic and the model architecture and parameters are both missing.

- Formulation needs to be clarified. The paper is focusing on the image-to-svg task, which is closer to reconstruction/vectorization instead of generation. There are many prior works on generation or conditional generation for SVG drawings, such as DeepSVG, as mentioned in the related works. Is there a particular reason to only focus on the reconstruction task?

**Questions:**

- It is not clear why the authors choose to use a VAE instead of applying the proposed loss terms to existing models.

- Parameter study can be improved by exploring wider parameter ranges (e.g., how does a very large/small weight affect the results?).

- Have the authors explored methods such as RLHF that can incorporate non-differentiable metrics?

- The detail of 4.2 is missing. What are the 'series of structured questions'?

- Will this proposed method work on more complex SVG drawings, such the ones generated by VectorFusion[1]?

[1] Jain, Ajay, Amber Xie, and Pieter Abbeel. "Vectorfusion: Text-to-svg by abstracting pixel-based diffusion models." Proceedings of the IEEE/CVF Conference on Computer Vision and Pattern Recognition. 2023.

---

### Meta-Review · Area_Chair_fexq · 2023-12-07

**Metareview:**

Most of the reviewers see the importance of the problem that the work is addressing. The AC agrees that the readability of the generated SVG code is important in addition to the visual quality. The authors made some nice progress by proposing new metrics and losses. However, the work suffers a few issues. First and foremost the paper lacks visual contents. As a SVG generation paper, it is hard to judge the quality of the generation without visual examples. Reviewer Zk7q pointed out that "No qualitative results or visualizations provided. It will not only help readers to better understand the criteria, but also demonstrate how the proposed method improve the SVG generation compared to the baseline methods." Reviewer z767 also mentioned the same issue: "For example, paper provides no visual results, making it very hard to judge the quality of the method." Second, the work lacks proper evaluations. when it comes to readability, it is important to conduct a user study to find out how human users react to the generations. The reviewers also pointed out the issues with the technical details of the work. For example, Reviewer p9Qj pointed out the design of some of the metrics is problematic. Finally, Reviewer p9Qj suggested a few additional analysis which could be nice improvements to the paper.

**Justification For Why Not Higher Score:**

The reviewers are generally concerned with the quality of the paper including both technical and presentation aspects.

**Justification For Why Not Lower Score:**

The authors intended to address an important problem that can be potentially impactful.

---

### Decision · Program_Chairs · 2024-01-16

Reject